# Navigating mental health needs at sea: Perspectives from the Canadian maritime community during the COVID-19 pandemic

**Hugo Andres Rojas Aldieri** ⓘ*, **Jennifer Shea, Shree Mulay, James Valcour** ⓘ, **Desai Shan** ⓘ

Division of Population Health and Applied Health Sciences, Faculty of Medicine, Memorial University of Newfoundland, St. John's, Newfoundland and Labrador, Canada

* arojasaldier@mun.ca

## Abstract

Seafarers play a vital yet often underrecognized role in sustaining global trade, facing unique mental health challenges, that was further exacerbated by the COVID-19 pandemic. Existing literature underscores elevated risks of anxiety, depression, and suicide among seafarers, heightened by isolation, extended contracts, and diminished shore leave. This is the first qualitative study exploring seafarer mental health needs and support use in Canadian waters. Guided by the research question—*What are the mental health needs of seafarers operating in Canadian waters during the COVID-19 pandemic, and what were the barriers that prevented them from accessing support?*—this study sought to identify how several socio-ecological factors shape seafarers' mental health needs and use of support. Thirteen interviews (eleven seafarers and two key informants) were conducted between February and October 2022. A hybrid thematic framework approach was adopted: an initial inductive thematic analysis generated data-driven codes, which were subsequently organized through framework analysis by deductively mapping them onto the domains of the Socio-Ecological Model. Interpretation was complemented by Ungar's social-ecological resilience perspective to interpret how individuals "navigate" available resources and "negotiate" for mental health support. Findings reveal that pandemic-driven mobility restrictions—such as denial of shore leave and repatriation challenges—intensified the pre-existing stressors of seafaring life. Inadequate onboard connectivity, reports of workplace harassment, and limited coordination among authorities, employers, and welfare organizations further constrained mental health support access. Port-based welfare centres, ship inspectors, and collaborative interventions demonstrated a potential to mitigate psychosocial risks. The pandemic spotlighted systematic gaps in the mental health supports available to seafarers in Canadian waters. Addressing these gaps requires multi-level strategies that link onboard, port-based, and policy-level supports. Adapting Mental Health First Aid and Stepped Care 2.0 to maritime

**Data availability statement:** The data are not publicly available. This study includes interview data for which the consent form explicitly stated that only the research team would have access to the data. In addition, even after de-identification, the full interview transcripts may carry a risk of indirect identification of key informants by informed readers. Memorial University of Newfoundland's Interdisciplinary Committee on Ethics in Human Research (ICEHR; ethics file 2022099-ME) has confirmed that, on this basis, requests for access to these data would be denied. Questions regarding these restrictions may be directed to ICEHR, Memorial University of Newfoundland, icehr@mun.ca, Tel: +1 709-864-2861. A limited exception may apply only where the requesting researcher has obtained prior approval from the relevant Research Ethics Board for secondary use of data.

**Funding:** This work was supported by the Canadian Institutes of Health Research (Operating Grant: Emerging COVID-19 Research Gaps and Priorities – Differential Impact of COVID-19 on Historically Excluded or Underserved Populations, Funding Reference Number EG2-179434 to DS) and by Memorial University of Newfoundland (Seed, Bridge, and Multidisciplinary Fund to HR). The funders had no role in study design, data collection and analysis, decision to publish, or preparation of the manuscript.

**Competing interests:** The authors have declared that no competing interests exist.

settings, coupled with stronger interagency collaboration, can enhance timely, context-specific mental health care.

---

# 1. Introduction

## 1.1. Research problem and gap

Despite seafarers being essential to Canadian trade, their mental health needs and access to mental health support remain understudied [1]. This gap underscores the need to understand the unique mental health vulnerabilities of seafarers. Working under prolonged isolation conditions, concerns about families, and eroding shore leave practices, links prior research on seafarers with elevated risks of anxiety, depression, and suicide [2–7]. The nature of seafaring work—characterized by extended periods at sea—was exacerbated during the COVID-19 pandemic [8–13].

Seafarers face significant challenges in accessing timely mental health support due to the inherent isolation of life at sea and numerous logistical barriers—such as limited internet access and physical isolation—that hinder access to professional mental health care, as highlighted by Lefkowitz and Slade (2019) [10]. In addition, barriers to reliable onboard support arising from the unique operational constraints of maritime work further obstruct effective mental health support [14]. Emerging evidence indicates that information and communication technology-enabled interventions—such as tele-counselling, mental health mobile apps, and helplines—may offer promising solutions to overcome traditional access barriers. For instance, Abila et al. (2023) argued that technology-based initiatives can meaningfully enhance seafarers' connection to mental health support, particularly when they yield benefits that directly extend to their families [15].

## 1.2. Policy context

In Canada, task shifting, Mental Health First Aid (MHFA), and Stepped Care 2.0 (SC 2.0) emerge as potential strategies to support workers at sea. "Task shifting" redistributes mental health tasks from specialists to trained non-specialists, boosting community capacity and enabling faster, basic care [16–19]. In this context, MHFA is a training program developed by the Mental Health Commission of Canada which decreases stigma and builds lay persons' capacity to recognize and respond to mental health issues [20,21]. SC 2.0 is an accessible service delivery model that offers a framework to organize mental health care by mapping existing resources onto a 9-step framework. SC 2.0 has been widely implemented in Canada, dramatically reducing wait times for mental health care in Newfoundland and Labrador and the Northern Territories [22,23].

## 1.3. Research question

The research question guiding this study is: What were the mental health needs of seafarers operating in Canadian waters during the COVID-19 pandemic, and what

barriers prevented them from accessing support? This question is critical because understanding these needs can inform targeted policy interventions to improve the well-being of seafarers.

## 2. Materials and methods

### 2.1. Ethics statement

Informed consent procedures, interview guides, and the proposal for this research have been reviewed by the Interdisciplinary Committee on Ethics in Human Research (ICEHR) and found to be in compliance with Memorial University's ethics policy.

Written consent was waived to protect participant anonymity. Instead, verbal informed consent was obtained prior to initiating the recording. Participants received a copy of the consent form, which was read aloud by the interviewer. Questions were invited and addressed before consent was requested.

Consent was documented through an audio-recorded statement made by the interviewer at the start of the recording, confirming that the consent form had been read and explained, that the participant appeared to understand its contents, and that consent had been granted. This verbal consent procedure was reviewed and approved by ICEHR.

### 2.2. Participants

This research is part of a wider community-level mixed methods study. Recruitment for seafarer interviews used three pathways, (1) email communications distributed by union partner organizations to their membership, (2) an invitation embedded at the end of the study's associated survey, where respondents could indicate interest in a follow-up interview by submitting contact details, and (3) social media posts on seafarer-specific channels. Following initial recruitment, purposive sampling was used to identify key informants with in-depth insights on seafarer mental health needs.

Because recruitment occurred through indirect channels, the research team could not determine how many individuals viewed the invitation, considered participation, or actively declined. Individuals who contacted the research team to volunteer were provided with study information and offered scheduling options. All participants who scheduled an interview attended as planned. No participants who scheduled an interview explicitly withdrew after consent.

In alignment with the epistemological stance of the broader project, the notion of pragmatic saturation was employed to define the sample size rather than relying on theoretical saturation. This stance focuses on whether additional interviews yield substantively new, practicable insights related to our research questions [24–29]. Empirical evidence suggests that qualitative studies may achieve theoretical saturation with sample sizes ranging from 10 to 20 participants, though the exact number depends on the richness of the data [24,30,31]. Thirteen interviews are described in this paper: 11 seafarers working in Canadian waters during COVID-19, plus two key informants (Table 1).

### 2.3. Research process

Interview guides for seafarers and key informants included introduction questions to understand the participants' work, central questions exploring the impact of COVID-19 on their work life and their mental health, and forward-looking closing questions on priorities and recommendations to address seafarers' mental health needs and barriers to accessing support at sea.

The seafarer interviews were largely conducted remotely (via Google Meet, Zoom, or phone) and occurred either onboard ship or at home while on leave between contracts, between February 1st to October 30th, 2022, with one asynchronous email interview. Key informant interviews were conducted in person at their workplaces. Transcriptions were made via Otter.ai and supplemented with field notes. NVivo (version 15) was used for data management.

### 2.4. Researcher positionality and reflexivity

All interviews were conducted by the first author (HR). At the time of the study, HR held an MD, MSc, and Postgraduate Diploma, and was a Master's student in transition to a PhD programme at Memorial University of Newfoundland (MUN). In

**Table 1. Interview participants.**

| Pseudonym | Description |
| --- | --- |
| Daniel | Daniel is a 25-year-old male from Ontario with 3 years of seafaring experience. He serves as an overseer of daily operations on a bulk carrier. During the pandemic, he continued to work in Canadian waters. |
| Michael | Michael is a 26-year-old male originally from Africa, now residing in Ontario. He has 8 years of seafaring experience, 5 of which occurred in Canada, performing duties as an ordinary seaman on a cargo ship. During the pandemic, he sailed both internationally and in Canadian waters. |
| Jacob | Jacob is a 26-year-old male from the United States who has worked as a second mate for 6 years. Serving on a carrier, he operated in Canadian waters during the pandemic. |
| Matthew | Matthew is a male seafarer from the United States with about 4 years of experience. Employed as a technician on a cargo ship, he handles navigation and repair-related tasks. He worked in Canadian waters throughout the pandemic. |
| Brandon | Brandon is a male seafarer with 7 years of experience as a supervisor on cargo vessels. He divides his time between at-sea duties and onshore tasks, travelling internationally and through Canadian routes during the pandemic. |
| Adrian | Adrian is a male from Ontario with roughly 2 years of experience. Occupying a support-staff role on a cargo ship, he operated in Canadian waters throughout the pandemic. |
| Nathan | Nathan is a male from Ontario with 7 years of experience at sea. He works as the director of operations on a cargo vessel. During the pandemic, he remained active in Canadian waters. |
| Genevie | Genevie is a seasoned female seafarer working as a cook. She has served on cargo vessels on the Great Lakes and coastal routes. During the pandemic, she took on relief assignments involving Canadian waters. |
| Andrew | Andrew is a Canadian watchkeeping engineer working on a drilling platform. With a decade of experience, he has worked on various vessel types and has operated on the Great Lakes, the St. Lawrence Seaway, and other Canadian routes. |
| Kevin | Kevin is a male seafarer working as a cook on an oil tanker across different sea regions. He has remained active on voyages involving Canadian waters throughout the pandemic. |
| Paul | Paul is a Canadian master mariner living in Newfoundland who alternates between Arctic operations in summer and international routes in winter. He continued commanding vessels through Canadian waters during the pandemic. |
| Grace (Key Informant) | Grace is the primary coordinator at a seafarer welfare center located at a major Canadian port, in charge of volunteer training, ship visits, and liaising with ship captains and local authorities. |
| Samuel (Key Informant) | Samuel is a ship inspector for a major Canadian port, responsible for overseeing maritime labour standards and seafarer welfare. Samuel collaborates closely with Canadian authorities and unions to uphold seafarers' rights both domestically and internationally. |

Note: All names are pseudonyms assigned by the researcher to protect participant confidentiality.

addition to his role as a graduate student, HR worked as a consultant in instructional design and knowledge mobilisation. He identifies as a man.

HR had formal coursework exposure to qualitative methods during graduate training, including interview design and thematic analysis. While he had not previously led independent qualitative research projects, he brought transferable skills from clinical practice, including experience conducting patient interviews, managing sensitive disclosures, and navigating professional boundaries in health-related conversations.

No prior personal or professional relationship existed between HR and the seafarers or key informants before recruitment. Some key informants had an existing professional relationship with a co-author (DS); however, HR had no prior contact with participants before data collection commenced.

At recruitment and consent, participants were informed that HR was a graduate student researcher at MUN with a clinical background. They were explicitly told that the research was conducted independently from employers, regulators, unions, and welfare organisations, that participation was voluntary, and that responses would be anonymized and not shared in identifiable form. The purpose of the study—to explore mental health needs and barriers to care among seafarers—was also explained.

This manuscript is part of a wider mixed-methods study. HR approached the study as both a clinician trained within a biomedical framework and as an outsider to maritime labour and seafaring contexts. In positionality terms, HR occupied an "outsider" position to seafaring work and maritime labour relations, while holding a "quasi-insider" perspective on mental health through clinical training. HR also brought lived experience of mental health challenges, which was treated as a sensitizing perspective rather than a source of authority. Early in the research process, reflexive consideration identified a tendency to foreground individual-level explanations, such as screening and clinical access barriers. To mitigate this orientation, the study design intentionally incorporated qualitative methods and a socio-ecological framework to privilege participants' lived experiences and broader structural determinants.

HR was also mindful that his clinical and academic status could create perceived power differentials, particularly when interviewing seafarers discussing workplace vulnerabilities. To reduce these dynamics, he emphasised independence from regulatory and employment structures and reiterated confidentiality protections during interviews.

Reflexive field notes were maintained throughout data collection and analysis to document evolving assumptions, contextual influences, and interpretive decisions. The interview guide was reviewed with the supervisory committee and industry partners prior to data collection. During analysis, peer debriefing was used to examine emerging codes and themes, providing an additional reflexive check on how meaning was co-constructed and interpreted. These practices were implemented to enhance transparency, credibility, and alignment with qualitative reporting standards.

## 2.5. Methods

A hybrid thematic and framework analysis was employed [32–34]. Moreover, peer debriefing—a process of discussing emerging findings and interpretations with academic peers to enhance reflexivity and analytical rigour—was employed. The primary interpretive strategy was thematic analysis, an inductive assessment that allowed insights to emerge directly from participants' accounts. Framework analysis was integrated as a complementary, deductive strategy by mapping identified themes onto the five dimensions of the socio-ecological model (SEM) [35]. The SEM was chosen because it offers a robust lens for understanding the complex, multi-layered factors that shape seafarers' mental health needs by situating individual experiences within the broader context of interpersonal, organizational, and societal influences, reflecting the deep connection between life at sea and mental health. Data was further interpreted through Ungar's social-ecological resilience perspective [36]. This model emphasizes that resilience emerges through dynamic processes by which individuals, families, and communities navigate available resources and negotiate for resource utilization, all within specific cultural and environmental contexts.

## 3. Results

### 3.1. Individual level

#### 3.1.1. Impact of the pandemic on mental health.
Participants described the impact of the pandemic on their mental health; some highlighted that *"mental health has always been an issue with seafarers"* (Grace, welfare center coordinator). Andrew, an engineer on a drilling platform, echoed that for some, harsh conditions existed before the pandemic, especially

around fatigue and working schedules, stating that *"nothing has changed. It may have for others, but at sea, you are either working or resting."*

Fear of COVID-19 infection was prevalent onboard ships but also for families at home, as Adrian explained, *"So, so much stress…. What if they get COVID-19?"* Adrian also explained the toll of these issues on individuals' mental health, *"you get fatigued… easily frustrated and angered. I saw my [crewmate] getting depressed."* Brandon, a supervisor on a cargo ship, echoed the challenges around fear of COVID-19, *"It has been quite challenging, we were scared of all our families, we're also scared to go to work because of the danger that was outside."*

Furthermore, factors operating at higher socio-ecological levels—such as public health policies, extended service on ships, or interpersonal issues like separation from spouses and families—exacerbated mental health challenges at the individual level during the COVID-19 pandemic. Brandon said, *"My management had accommodated us in the hotel… I was depressed because I was not close to my family. And I was also going through some anxiety… it affected me quite a lot."* Matthew also referred to mental health impacts due to increased isolation from families, compounded by communication restrictions, *"[between July to October 2021] it was very, very demanding, we weren't even allowed to make random calls to our family… actually underwent some anxiety and depression without talking to my family."* Similarly, Jacob, a 2nd mate, referred to mental health issues arising from communication restrictions during 2020 and 2021, *"My mental health was quite disturbed because communication between my fiancée and me was not functioning well. So, it gave me a lot of sleepless nights. I was stressed, quite depressed."*

Denials of shore leave and repatriation, and the need for testing and vaccination, deeply impacted seafarers' mental health. Grace noted, *"…not only is mental health always an issue… COVID-19 caused all these contract extensions... they couldn't go home."* Moreover, denial of shore leave, also reduced protective mental health factors, such as enjoyment of the job and leisure time. For instance, Genevie referred, *"During the height of the infections, we weren't allowed shore time, that was rough."* Paul, a captain, indicated that *"One of the joys of this career is being able to see the world. That's sort of how it's marketed. It's pretty much been zero since the pandemic was first announced.* Similarly, Samuel, a ship inspector, said,

> *"Seafarers are isolated enough as it is, but during the pandemic, they could see the city right in front of them, but weren't allowed to leave the ship. It was frustrating for them. Just imagine your ship sailing right through the center of the city... and you can see everything that's going on, the hustle and bustle of the city."*

The need for COVID-19 testing certification, isolation, and vaccination also brought new challenges for essential workers, such as extended processing time in ports, further reducing opportunities for rest and leisure. Nathan, a director of operations on a cargo ship, explained, "We had to cross-check the goods and also get a *COVID-19-free certification, so we took a lot longer than usual*." Paul reflected that if seafarers *"had to go home and isolate for two weeks every time we got off the ship, our free time at home would be seriously curtailed."* Michael shared, *"So you have to produce the certificate and then for nations to accept it. Otherwise, we will be forced to undertake tests, which, especially in Africa, may take weeks. So, so tiring, so time-consuming, more stressful, more fatigued."* A reduction in workers further compounded these challenges, Nathan further explains, *"some members were quarantined. It really affected me because sometimes we couldn't sleep well because we had a lot of things to do, we didn't have breaks… So, it was really affecting mentally*." Brandon echoed*, "I would say there was a reduction of the amount of staff present on the ship… most of us had to do extra work."* Andrew elaborated, *"when people do get sick, crewing usually has to go up through the list and try to replace such person, however the lists are often times without spares as companies reduce crews as much as possible."*

Discrimination around COVID-19 also impacted essential workers' well-being by causing stress and frustration. Michael recounted, *"I remember in Africa and Europe, there was so much discrimination, and the citizens looked at you*

*suspiciously."* Kevin, a cook, shared, *"You go into a port, and you have to ask permission to take a walk. And you really feel like a child being watched."*

Examples of mental health impacts of COVID-19 included experiences of anxiety, depression, fatigue, and fears of COVID-19 for themselves and their families. Prolonged isolation—denied shore leave, contract extensions and communication restrictions—undermined seafarers' mental health, while reduced manning, mandatory testing, and quarantine requirements increased fatigue and stress. Experiences of discrimination ashore further compounded their frustration and sense of alienation.

**3.1.2. Negotiation of mental health resources.** Participants expressed varied perceived sense of importance and self-efficacy when it came to managing mental health challenges. For some, like Genevie, mental health is placed as a higher priority than staying on the job itself. She shared, *"nothing's worth your mental health, quit your job, walk away. That's always my advice."* Nonetheless, she expressed little self-efficacy in getting mental health support at sea when needed:

> *"The only thing you can do is wait till you're in a port and go to the emergency room, that's about it. Or you just say I'm getting off at the next port. But, that's easier said than done. Because you might not be in a financial position to be able to do that."*

Brandon showed a positive attitude towards talking about mental health and suggesting the existence of spheres where mental health concerns are openly talked about, saying *"I think there's every reason for everybody to always be open to sharing the state of welfare. And I'll say everybody in total that I know about 90%, they're always open to sharing the affairs of their mental health."* Andrew, on the other hand, showed a preference to keep mental health issues to himself, he explained his preference by stating, *"I don't like to bother others with my issues… I like having control over my own situations… when people you know and work with get their hands into what you got going on, I find things turn into a tangle."*

These reflections reveal three core needs. First, seafarers require practical, readily accessible support at sea, not just waiting to get off the ship, but onboard services (e.g., tele-counselling or emergency response) that overcome industry barriers. Second, they benefit from a culture of peer openness**,** where talking about mental health is encouraged and normalized. Third, they also need confidential, self-directed resources that respect individual autonomy and privacy for those who prefer to manage distress privately.

## 3.2. Interpersonal level

**3.2.1. Interpersonal factors impacting seafarers' mental health needs.** Interpersonal relationships on board and family relationships were highlighted as key factors serving both as a component of positive mental health as well as a protective factor against mental stress, strain, and illness. The role of family and peer relationships as a protective factor can be evidenced by participant accounts such as Nathan's: "I prefer talking to a therapist or my family member, for reasons *that I trust my family."* Paul elaborated on the role of friends and families as informal sources of mental health support:

> *"The mental health help I do get is 100% friends, co-workers, and family. I lean quite heavily on one of my colleagues… It's very nice to have a friend on board. I call home a lot, and having good communication is essential for that."*

Others echoed the value of on-board confidants to navigate the stress associated with their profession. For instance, Brandon stated, "My best friend during that period was like a therapist to me because he helped motivate me, *it gave me a reason to stay strong, to move on."*

**3.2.2. Negotiation of interpersonal support.** Close relationships, such as onboard friends or close co-workers, and higher-ranking roles are highlighted as sources of interpersonal support. Kevin elaborated on the importance of approachable captains, *"[my] relationship with [my current] captain is excellent, he is a young captain really focused on communication… I can tell him anything..."*

Genevie's experiences also reflect negotiation by seeking the person most likely to be welcoming and with sufficient agency to provide support:

*"If you've got a good first mate, that's definitely the person you want to approach. But you got horrible first mates too, in which case you find somebody else to talk to, but the captains don't tend to want to be bothered with stuff like this…. Second mate and third mate, they have no authority… if you're in the engineering department, you would go to the chief engineer."*

Camaraderie was also evidenced as a resource for negotiating further types of support. Samuel, a ship inspector, shared a story of seafarers grouping together during adversity to collectively demand an appropriate response from ship management following a traumatic event:

*"We had seafarers who got hit by a wave, they were trying to stop the ship from sinking, and one of them died. So, there's a seafarer in the freezer… then the four seafarers retreated to the bosun's room, shut the door, and told the captain, we're done. Because first, you don't want to go back to work after something like that. You want to go home."*

Samuel's account illustrates that, when formal support is absent, seafarers may come together and take collective action—barricading themselves and demanding to go home—to ensure their needs are met. It underscores both the depth of their distress after a traumatic event and the necessity of having clear, accessible protocols for post-incident care on board.

## 3.3. Community level

**3.3.1. The impact of shipboard culture on mental health needs.** Beyond higher-ranking officials and close personal relationships, the overall crew's community and culture were reported as a determinant of onboard well-being, where issues such as harassment and stigma pose a significant threat to individuals' mental health. For instance, Genevie shared, *"Some boats are just toxic. In one instance, the ship had posters all over the place, you know, their harassment policies… and then they continue to allow this, this chief cook to terrorize people."* Mental health stigma highlighted as a factor impacting mental health experiences onboard, as shared by Samuel, *"there's also the stigma that comes with mental health… these guys, they don't want to show any weakness."*

On the positive aspects of the ship-based community, the interviews demonstrated that for some crews, captains play a central role in establishing positive mental health spaces. Paul, a captain, expressed the importance of the mental health of his crew. However, he did express limitations on the type of mental health support he is able to provide, while noting an increased need for this type of help since the pandemic:

*"So, I very much run on an informal; the door is always open... And I hadn't had anyone take me up on it, other than minor issues, until the last two years… It's been interesting because I don't really feel equipped to deal with some of the issues that have come forward."*

**3.3.2. Port-based community, as a protective mental health factor for seafarers.** Ship inspectors, welfare facilities, and chaplains play an essential role in facilitating the interface between seafarers and land-based services. Samuel exemplifies the role of inspectors by explaining their response to instances where seafarers were found to be working under forced contract extensions, stating: *"Ideally, we get the authorities to detain the vessel. [other] times we do a flag state approved repatriation plan, which means they're going to be repatriated from the next port."* While

acknowledging the limitations of the type of support available to seafarers at sea, Samuel also shared mental health resources available to them through ship inspectors: "*a lot of us inspectors, we have, like online tools, self-diagnosis tools and stuff that we can give to seafarers as a resource, but, you know, that's not ideal.*"

Moreover, acknowledging the deep concern of seafarers for their families, the interface between some ship inspectors and unions demonstrated an example of protecting seafarers' mental health by providing holistic support to families at home. Samuel elaborated on it by saying,

*"The model we use is that unions are like families. We have family training that goes on… where they learn to under-stand what your parents are doing and why they're doing that to support your family… that makes, makes the world make a little more sense to you know, the young kid who's waiting for his dad, who's never home, and the wife that's always lonely because his husband's always working right."*

Similarly, port-based welfare facilities proved themselves essential, especially for the most vulnerable seafarers, as illustrated by a story of Grace supporting a seafarer in distress:

*"I was home on a Sunday afternoon and received an urgent phone call from a seafarer who was in the Atlantic Ocean beyond Newfoundland. His baby brother was threatening to commit suicide. This man needed somebody to talk to, to help him through this. I basically communicate with him on the phone and through email most of the day. Meanwhile, trying to get a psychologist to get on the phone with him, I've had a lot of training, but I'm not a psychologist, but just the fact that we communicated constantly for about five hours that day made all the difference in the world to that man."*

Welfare facilities also exemplified the importance of proactive mental health support, acknowledging that sometimes seafarers may not seek help even when needed. Grace shared,

*"We try very hard to ask all the right questions when we go on the ships… [when you're a trained person] you can tell that there's something wrong. So, I would go and ask questions or get a ship inspector to find out what the problem is and detain a vessel if it needs to be detained."*

**3.3.3. COVID-19 impact on the port-based maritime community.** Notably, COVID-19 posed a major challenge to sustaining the support provided by the port-based community. Samuel, a ship inspector, shared: *"The pandemic made everything harder... So, I had to figure out how to become more accessible without being on the ship… it impacted the number of vessels I could visit, impacted how many seafarers I would see."*

Grace, a welfare center coordinator, described the impact of COVID-19, such as a reduction in volunteers and quality of interaction with seafarers during ship visits: *"We don't have as many volunteers volunteering because of COVID-19."* She also shared:

*"[seafarers] haven't been able to get off the ship and if we can't get on the ship and go into the mess and sit for 20, 30, 40, minutes, then you're not getting the information that you need to make sure that the seafarers' welfare is as good as it can be, right?*

COVID-19 also impacted the funding sources of port-based facilities, as Grace shared,

*"We have really no funding. That's one of our greatest challenges, especially during COVID-19, we couldn't have our tournament or any of our dinners or luncheons. So, I have been begging our ongoing sponsors, even though we're not having an event, to please support us anyway so that our seafarers need us more than ever before."*

**3.3.4. Negotiation of support at the community level.** Port-based welfare staff navigated and negotiated resources on behalf of seafarers. Grace exemplifies the role of welfare workers navigating barriers to accessing health care on behalf of seafarers:

*"Well, I wasn't going to go to the emergency room because I knew that we'd be sitting there for six or seven hours and not getting anywhere at all. So what I did is I put them in the car, and I took them to my ophthalmologist, and we got him in, and he had a piece of steel in his eyeball, so they took it out and gave him a prescription, a couple of prescriptions, one for pain and one for infection, and we were in and out of there within an hour."*

One captain's extreme story provides a further example of negotiation of support on behalf of seafarers:

*"The captain had been on that vessel for two years. During those two years, his*

*mother died, and his wife divorced him. He said, "I have not been off the ship for two years. I need your help. I am no longer capable of being the master of this vessel. I need a replacement, and I need it now. Can you help me?"*

Ultimately, the port-based community, led to authorities intervening. As Grace explained, *"I harassed that shipping company… a captain came over and relieved him. He went home. He had an eight-year-old daughter he hadn't seen for over two years."*

Collaboration between ship inspectors and welfare staff is constant, and other instances of collaboration were evidenced during the interviews. For example, welfare facilities partner with medical service providers to train their staff to conduct basic general health assessments in collaboration with a telehealth medical service provider to facilitate access to health and mental health services for seafarers on ships and ports. Grace shared,

*"[provider] they're emergency specialists... So basically, you interview the individual, take all vital statistics, find out what the problem is… make a telemedical call, and talk to a doctor on the phone, and they will come and look after you, or they will send us a prescription…"*

Grace also explained her response choices when additional support is needed: *"I send the information through to our ITF representative. And then I also have some really great connections with TC."* Similarly, Samuel shared an example of a case that needed the involvement of diplomats to resolve a case of forced contract extensions:

*"That case was tough. This ship with a Singapore flag, seven-month contracts extended month by month until 13 months. Burma closed its borders, which meant we had to bring in the Canadian diplomats to contact Burma and say, you know, you must open your borders to your people and allow them to come back."*

### 3.4. Organizational level

**3.4.1. Organizational factors impacting individuals' mental health needs.** Connectivity issues, stigma, abuse of power, neglect of essential needs, and organizational culture (different from ship-based culture) were highlighted as organizational factors impacting seafarers' mental health needs. Connectivity issues were prevalent. Genevie shared, *"Oh, it's terrible. Terrible. Yeah, it's really hard… sometimes in Lake Superior, you might even lose email."* Similarly, Nathan expressed, *"the main barrier is communication… sometimes you need to talk to a mental health therapist, but there's no connection, there's no network."* On the other hand, good connectivity can act as a protective factor by enabling leisure time and communication with families, as indicated by Brandon, *"the internet is great… very high level. So, I would say*

*it has been great. Most of the time, we try to do FaceTime with my family. It has been successful."* Beyond the internet, regular phone coverage also presents a barrier to getting help, as Andrew explains:

*"There is a 1-800 number that the company has provided usually but if you are 200 plus miles offshore, what cell service provider is going to be having coverage?... and I believe some providers for this service are quite expensive... and it's not something that you would want to be trying to reach out on the satellite phone where everyone is listening."*

The cost of satellite phones was also mentioned as a barrier by Andrew, "Satellite phones could be quite expensive, and some companies would like to refrain from using them*, unless it's an absolute emergency, and this is another reason why it is often frowned upon."*

Harassment and abuse of power play a significant role in individuals' mental health, which may be determined at the organizational level through organizational culture and enforcement of harassment policies. For instance, Genevie shared,

*"The companies are all talk about zero tolerance for harassment… But my objection was that over the years, they're just some horrible people… so horrible… just tyrants… bullies… And the companies, the union, everybody knew who they were, but nobody did anything about it. It was tolerated by the companies and the union. And that was really hard. And now finally [company A] did fire one of those cooks, but then [company B] hired them back… the companies protect these people."*

Another impactful story was shared by Andrew, depicting abuse of power by employers using seafarers' employment as a contingency to make unreasonable demands:

*"An old colleague was in the Delivery Room; his wife was in labour with their firstborn… His phone started ringing [the crewing department] said they needed him to join the ship on that same day, saying, 'Well, you aren't the one in labour, so will you join?'"*

Upon rejecting the request, the seafarer found his position under review, even when he had processed paternity leave months prior.

The pandemic heavily influenced the neglect of essential needs. Milder cases, such as those experienced by Genevie, included limitations on fresh vegetables and fruits and a reduction of food variety onboard. In a more extreme case, Nathan shared:

*We needed to work longer than usual. So, the food supplies that we got for that period of time weren't sufficient… we ran out of supplies, it was really bad, we fed on [improvised] seafoods."*

Grace's story provides another account of seafarers fishing due to a lack of food supplies:

*"A month ago, I got a really bad feeling, because seafarers were fishing off a vessel, and when we asked, Why are you fishing? Well, we have barely any food. We don't pick up our provisions for another two weeks, and we don't have enough to cook."*

Additionally, the onboard environment for leisure, physical activity, and recreation was another important factor impacting seafarers' mental health needs. Nathan shared, *"Sometimes we work very tediously, and we don't really get much time to interact, or to ease up, pray, the workload is always too much."* Availability of leisure and physical activity spaces seems

to be a protective factor, as expressed by Jacob, "we have a little room where we can actually walk, do some push-ups, sit-ups, to try and keep ourselves fit. This has helped mentally."

### 3.4.2. Navigating and negotiating occupational resources.

Access to health services commonly occurred through organizational channels, though marked by geographical constraints. In some cases, proactive mental health outreach represented a protective factor, as shared by Nathan, "the [therapist] doesn't actually come with us onshore, but once in a while the person calls to find out if we are okay." Similarly, Brandon shared, "My company has counsellors who always provide mental health support to seafarers who are working on board." He also said, "There is always a case for seafarers to see a therapist before embarking on any journey. Because it's something that my company has had goals in place for about four years now, and it has been working for everybody."

While other participants did not experience such proactive mental health follow-up, some of them echoed the need for "on-board medical practitioners giving regular checkups"(Matthew), referring to "any option on board as a *big improvement*" (Paul).

Negotiation was also relevant to telehealth, Paul also shared:

"There are a few phone-based services, but I think mental health is so difficult to treat over the phone. I think it's, you know, that personal relationships are such an important part of it. I don't feel the quality is very high."

Genevie demonstrated how seafarers navigate connectivity issues, negotiating personal time to achieve effective communication: "The *internet in the evening is so horrible… if I need to send emails, I do it at five o'clock in the morning because then you've got more bandwidth to work with."* Kevin also shared methods to adapt to communications restrictions in some waters:

"For communication between Newfoundland and Montreal, my cell phone is in the window… unless you'd use your own data, where data plans are also an issue… [my phone company] did not cover the United States. So, I will have to pay… the fees are ridiculous… I'd like to be able to speak to someone, but I'm not going to pay $20 a day every day I'm in the United States."

Seafarers also negotiate psychologically safe working conditions, while acknowledging barriers to achieving it, as Genevie states:

"What would they do for you?... You could say, Oh, the chief cook is picking on me. What are they going to do? I remember going to the captain, who said, "Oh, she's like that with everybody. Don't take it personally." Good grief... You know, firing that person and not allowing them to terrorize anybody ever again would have been the only thing that would have satisfied me."

In some cases, seafarers rely on support from port-based communities to negotiate a resolution to instances of harassment, yet they sometimes experience retaliation upon making complaints, as shared by Grace:

"Officers were harassing a crew, up all night drinking in the captain's cabin. So, I called ITF and TC… but then nobody wanted to speak up because they're afraid, intimidated by these bullying officers. Then, I got a letter with all the crew members' signatures and presented it. But unfortunately, the Cook, who was complaining, was still in his probationary period, so they let him go, and there wasn't anything we could do about it…"

Negotiation was even present at the moment of accessing physical exercise facilities, as shared by Brandon, "Yes, there are spaces, but I like to do my exercises in my free time. So, I always engage in my room."

## 3.5. Societal level

**3.5.1. Societal factors impacting seafarer mental health needs.** The COVID-19 pandemic itself constituted a major societal factor impacting mental health needs, representing an unprecedented crisis for seafarers, bringing along restrictions on essential needs such as shore time, timely repatriation, time at home, and, in some cases, even food supplies. Even issues distant from mental health, such as vaccination, represented a source of stress due to varying regulations, requirements, and vaccine availability at different ports, placing bureaucratic challenges as a critical concern for seafarers who depend on visas and travel requirements to join their work and return home. Samuel summarizes the COVID-19 period as:

*"One of the most difficult times seafarers have faced in modern history… the world forgot about them… everybody was so caught up in domestic problems that they forgot about the seafarers that were keeping the supply chain going."*

Fragmented governance, which refers to the limited coordination and overlapping responsibilities among federal, provincial, and maritime authorities, is a pre-existing pandemic issue, exacerbated by evolving regulations during COVID-19. The impact of fragmented governance on mental health can be explained through issues distant to mental health, such as COVID-19 vaccination access, causing stress for a mobile workforce amid varying mobility regulations. Grace shared:

*"I had to jump hoops to get permission from the Department of Health to allow seafarers to be vaccinated… So finally, one day, we brought a group of seafarers to a clinic, and all hell broke loose. Within 15 minutes, the Deputy Minister of Health wanted to have a Zoom meeting with me, along with the heads of pharmacies and the heads of other medical associations, and they said, "You just can't do that."*

Furthermore, the impact of societal isolation and marginalization of seafarers was summarized by Samuel as follows:

*"Well, I don't know if it comes from the pandemic, or if it just comes from my experience with working with seafarers, but I've learned that these guys are isolated and very marginalized.? When you put those two things together, it can make people feel unworthy or not valued."*

**3.5.2. Navigation and negotiation of societal factors influencing access to mental health support.** Fragmented governance demanded that maritime stakeholders engage in multilateral collaboration on behalf of seafarers, as exemplified by efforts of welfare staff and ship inspectors to negotiate with federal and provincial authorities to address emerging COVID-19 challenges, such as access to vaccines, health services, and essential supplies. Samuel explains the complexity of coordinating mental health support for seafarers by sharing,

*"Ideally, under the MLC, we're supposed to have one point of contact with the government, which would be TC. But once you get into health issues, you start dealing with regional health authorities… if you can't get them to detain ships for, you know, wages and guys that haven't been repatriated, it becomes a much bigger challenge when you're talking about a mental health issue, right?"*

## 4. Discussion

This study set out to explore the multifaceted mental health needs of seafarers operating in Canadian waters during the COVID-19 pandemic, identifying barriers that obstruct access to mental health support. By applying a hybrid thematic and framework analysis through the lens of the SEM and Ungar's social-ecological resilience perspective, we uncovered how individual, interpersonal, organizational, community, and societal factors interact to heighten or mitigate seafarers' psychosocial risks and determine the use of support.

### 4.1. Central contributions and advancement of disciplinary understandings

A key contribution of this work lies in revealing the interplay among multiple levels of influence, particularly how macro-level conditions such as pandemic-related travel restrictions and fragmented governance cascade down to shape individual well-being. Pre-COVID-19 studies have long documented seafaring as a high-stress occupation [2–7]. Our research claims that the pandemic amplified pre-existing stressors (e.g., fragmented governance, prolonged contracts, reduced shore leave, fluctuating staffing). Additionally, by situating these findings within Ungar's resilience theory [36], this study highlights that seafarers' capacity to "navigate" available supports and "negotiate" for their utilization is shaped not only by personal coping strategies but also by institutional structures and cultural norms.

### 4.2. Comparisons with prior research

Our findings align with existing literature that documents persistent barriers to mental health support among seafarers, including inadequate connectivity, stigma, and lack of onboard psychological services [10,14,15]. They also resonate with studies emphasizing the pivotal role of interpersonal bonds as informal yet essential forms of care [9]. However, where some scholars highlight growing digital solutions (e.g., telemedicine, virtual counselling) [15]. Our interviews point to challenges in practice, illustrating how limited port access and insufficient shipboard connectivity curtail telehealth's full potential and contribute to feelings of isolation, frustration, and helplessness among seafarers. Seafarers described satellite-based communication as costly and, at times, insufficiently private for sensitive conversations, while limited bandwidth required strategic "off-peak" communication that can compete with rest. In this way, connectivity operates not only as a channel for support, but also as a structural determinant of whether remote care is experienced as acceptable and usable—consistent with international evidence that internet access and social support function as key protective factors for seafarer wellbeing during COVID-19 [37]. In elaborating on these patterns, this study extends current conversations by showing how shore-based support networks (e.g., welfare centres and ship inspectors) function as an essential community asset in bridging the gap between maritime isolation and specialized health services.

International research during the COVID-19 crew-change crisis similarly documents how prolonged time onboard, uncertainty about repatriation, and restricted shore leave can intensify psychological distress and fatigue among seafarers [8,9,14]. Quantitative evidence also points to the buffering role of onboard peer support, employer support, and reliable internet access as protective conditions during this period [37]. This study complements that international evidence by detailing the pathways through which restrictions translated into distress: shore-leave denial undermined meaningful breaks and recovery; administrative requirements (testing, certification, quarantine) extended port-time demands and reduced rest; and cumulative uncertainty amplified frustration and helplessness—particularly when seafarers could see shore communities but remained confined onboard.

A further point of alignment with international maritime scholarship concerns harassment and abuse of power as psychosocial hazards at sea. Structured reviews describe bullying and harassment as recurrent shipboard risks sustained by steep hierarchies, multinational crews, and weak reporting pathways [38]. Recent fleet-level evidence likewise indicates that workplace violence remains prevalent in merchant shipping [39]. This study extends that international work by showing how harassment can become a direct barrier to help-seeking: fear of retaliation and mistrust in internal complaint mechanisms can redirect disclosure away from organisational channels and toward external actors (e.g., welfare staff, inspectors) perceived as safer intermediaries. This reinforces the need for credible, confidential reporting routes and post-incident support pathways that are workable within shipboard power dynamics.

### 4.3. Implications for future research, policy, and practice

Findings indicate a clear need for integrated, multi-level interventions that blend telehealth services, port-based outreach, and onboard mental health literacy initiatives. Building from Canada's existing mental health infrastructure, SC 2.0 offers

a promising framework to streamline and triage care, using tiered levels of intervention that match individuals to the least-intensive appropriate service [22,23]. This approach could be adapted to maritime contexts, leveraging and strengthening existing community assets, emphasizing telemedical consultations, thereby ensuring continuity of mental health care throughout a seafarer's voyage.

Similarly, MHFA, already available in specialized modules for youth and Indigenous communities [20,21], can be tailored to shipping environments, equipping both crew and shore-based personnel with strategies to recognize psychological distress and facilitate early intervention.

Internationally, seafarer mental health support is increasingly scaffolded through transnational welfare infrastructure. For example, ISWAN operates SeafarerHelp as a free, confidential, multilingual helpline available 24/7 year-round and maintains a Seafarer Centre Directory to help crews locate port welfare services [40]. In parallel, the ITF provides a welfare-centre index to help seafarers identify port-based spaces offering advice, a supportive listener, and facilities to contact home [41]. The Mission to Seafarers similarly signposts these global resources and offers chaplaincy-based support pathways [42].

Within global mental health, WHO strategies emphasise expanding access through community-based, integrated models and the use of non-specialist supports and decision tools to narrow treatment gaps [43,44]. This study adds to that global body of work by showing how help-seeking frequently requires navigation (identifying appropriate entry points) and negotiation (advocacy with employers, port actors, and local/regional health authorities). In practice, welfare staff and ship inspectors appear to often function as de facto "task-shared" mental health navigators—bridging seafarers to local services, escalating concerns, and negotiating feasible care pathways. Building on this observation, Canadian service-delivery assets discussed here—such as Mental Health First Aid and Stepped Care 2.0—can be reframed as exportable implementation strategies: (i) strengthening lay responder roles for ship/port personnel, and (ii) mapping existing international welfare resources into tiered options aligned to stepped-care logic. This approach strengthens—rather than replaces—existing assets.

The centrality of port-based welfare actors in these findings also aligns with international maritime labour governance. The Maritime Labour Convention (MLC, 2006) emphasises seafarers' access to shore-based welfare facilities and encourages the establishment of welfare boards to review adequacy of services as industry needs change. However, empirical research highlights that port welfare provision is frequently overstretched and dependent on voluntary organisations, rendering coverage vulnerable to funding and access constraints—pressures that were accentuated during COVID-19 [45]. Recognising welfare staff and inspectors as practical "mental health navigators" therefore represents an internationally relevant implementation strategy.

## 4.4. Strengths and limitations

A notable strength of this study is the dual methodological approach (thematic plus framework analysis), which enabled a nuanced understanding of emergent themes while mapping them onto the SEM. Seafarers and key informants (welfare coordinators and inspectors) enriched the data by triangulating perspectives on structural barriers and everyday realities.

Nonetheless, several limitations merit caution. First, the sample size (thirteen participants) was sufficient under pragmatic saturation principles but may not capture the full diversity of seafaring roles and vessels. Second, because most interviews were conducted remotely—by phone, videoconference, or email—(apart from two key informant interviews held in person), the study may have excluded seafarers with minimal internet or phone access, potentially biasing the findings toward those with comparatively better connectivity. Third, because the study focused on Canadian waters, some results may not transfer seamlessly to jurisdictions with different labour laws, community assets, or health systems. Finally, member checking was not conducted, and participants did not review transcripts or thematic interpretations. While this limits opportunities for participant validation, credibility was enhanced through reflexive documentation and peer debriefing to assess coding and theme development.

## 5. Summary

Seafarers continue to occupy a paradoxical position as "essential yet overlooked" contributors to global trade, a reality laid bare by the COVID-19 pandemic. Through a socio-ecological and resilience-based lens, this study reveals the intricate web of stressors that converge at the individual, interpersonal, organizational, community, and societal levels. Importantly, it underscores the network of port-based resources available to support seafarers' mental well-being. Canadian maritime policy, if synergized with tested mental health assets like SC 2.0 and MHFA, stands poised to lead by example, offering transformative, evidence-based interventions that can reverberate globally.

## Acknowledgments

The authors express their deep gratitude to all the seafarers who participated in the study, as well as the Seafarers International Union of Canada, North America Maritime Ministry Association, Canadian Merchant Service Guild, Mission to Seafarers, International Longshore and Warehouse Union — Local 400, and International Transport Workers Federation for their contributions to the development of the interview guide and distribution of recruitment materials.

## Author contributions

**Conceptualization:** Hugo Andres Rojas Aldieri, Shree Mulay, James Valcour, Desai Shan.

**Formal analysis:** Hugo Andres Rojas Aldieri.

**Funding acquisition:** Desai Shan.

**Investigation:** Hugo Andres Rojas Aldieri, Desai Shan.

**Methodology:** Hugo Andres Rojas Aldieri, Jennifer Shea, Shree Mulay, James Valcour, Desai Shan.

**Project administration:** Desai Shan.

**Supervision:** Jennifer Shea, Shree Mulay, James Valcour, Desai Shan.

**Writing – original draft:** Hugo Andres Rojas Aldieri.

**Writing – review & editing:** Hugo Andres Rojas Aldieri, Jennifer Shea, Shree Mulay, James Valcour, Desai Shan.

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
