## [Decision Letter · Decision Letter 0]

30 Jan 2026

PGPH-D-25-03416

Navigating mental health needs at sea: Perspectives from the Canadian maritime community during the COVID-19 pandemic

Dear Dr. Rojas Aldieri,

Thank you for submitting your manuscript to PLOS Global Public Health. After careful consideration, we feel that it has merit but does not fully meet PLOS Global Public Health’s publication criteria as it currently stands. Therefore, we invite you to submit a revised version of the manuscript that addresses the points raised during the review process.

The mental health of seafarers is an important issue that does not receive enough attention. Research in this area is therefore very welcome. However, this manuscript has several important shortcomings, many of which have already been clearly identified by the reviewer. In particular, the discussion section would benefit from deeper engagement with the findings and a clearer connection to existing international research on maritime mental health.

We look forward to receiving your revised manuscript.

Kind regards,

Susmita Chandramouleeshwaran

Academic Editor

Journal Requirements:

1. In the ethics statement in the Methods, you have specified that verbal consent was obtained. Please provide additional details regarding how this consent was documented and witnessed, and state whether this was approved by the IRB.

2. Please provide a detailed online Financial Disclosure statement. This is published with the article. It must therefore be completed in full sentences and contain the exact wording you wish to be published.

a) State the initials, alongside each funding source, of each author to receive each grant. For example: “This work was supported by the National Institutes of Health (####### to AM; ###### to CJ) and the National Science Foundation (###### to AM).”

For more information, please go to our submission guidelines:

https://journals.plos.org/globalpublichealth/s/submission-guidelines#loc-financial-disclosure-statement

3. Please ensure that the funders and grant numbers match between the Financial Disclosure field and the Funding Information tab in your submission form. Note that the funders must be provided in the same order in both places as well.

4. Please update your online Competing Interests statement. If you have no competing interests to declare, please state: “The authors have declared that no competing interests exist.”

5. In the online submission form, you indicated that "The data that support the findings of this study are available from the corresponding author upon reasonable request.".

a) In a public repository,

b) Within the manuscript itself, or

d) Uploaded as supplementary information.

For further assistance, you may go to: http://journals.plos.org/globalpublichealth/s/data-availability

Additional Editor Comments (if provided):

Reviewers' comments:

Reviewer's Responses to Questions

**Comments to the Author**

1. Does this manuscript meet PLOS Global Public Health’s publication criteria? Is the manuscript technically sound, and do the data support the conclusions? The manuscript must describe methodologically and ethically rigorous research with conclusions that are appropriately drawn based on the data presented.

Reviewer #1: Partly

2. Has the statistical analysis been performed appropriately and rigorously?

Reviewer #1: I don't know

3. Have the authors made all data underlying the findings in their manuscript fully available (please refer to the Data Availability Statement at the start of the manuscript PDF file)?

Reviewer #1: Yes

4. Is the manuscript presented in an intelligible fashion and written in standard English?

Reviewer #1: Yes

5. Review Comments to the Author

Reviewer #1: This study makes an important and timely contribution by examining the mental health needs of seafarers operating in Canadian waters during the COVID-19 pandemic—a population that is both essential and structurally marginalized.

However, limitations in reflexivity, positional transparency, and engagement with global mental health theory constrain the interpretive depth and transferability of the findings.

1. Insufficient reflexivity and researcher role specification: The study provides limited information on the researchers’ positionality, disciplinary background, or prior engagement with maritime or mental health sectors.

lack of clarity on researcher's positionality opens the door to researchers’ assumptions shaping the data collection or interpretation, lack of information n power dynamics in interviews (e.g., seafarers vs key informants), or how insider/outsider status influenced rapport and disclosure. In qualitative research, particularly with marginalized workers, reflexivity centered on researcher positionality is central to trustworthiness. Its absence weakens claims of analytical rigor.

Implication: Readers are left uncertain about how meaning was co-constructed and how interpretive decisions were made.

Address: this paper: Goundar, P. R. (2025). Researcher Positionality: Ways to Include it in a Qualitative Research Design. International Journal of Qualitative Methods, 24. https://doi-org.proxy.bib.uottawa.ca/10.1177/16094069251321251 is a good guide on how and where to include researcher positionality. This reviewer feels that including researcher positionality, how it has shaped the data collection and interpretation of the findings and the final conclusion is a necessary requirement in qualitative research and encourages the researchers to include in their paper. This point reflects need for a major revision. This reviewer recommends to the researchers to look at the COREQ domain 1 (research team and reflectivity) points and include all of them in their paper.

2. Points for minor revisions: a) participant recruitment pathways are insufficiently detailed (e.g., via employers, unions, welfare orgs).b) No discussion of refusals, dropouts, or reasons for non-participation. Important given potential fear of reprisal among seafarers. c) context of data collection (setting-where the participants on leave or on board for example) during the interview is critical for interpreting disclosure/power dynamics.

3. Limited engagement with global mental health research: This reviewer will encourage researchers to include a discussion point on how their current research engages with larger global mental research ( There is a mention on how it aligns globally (lines 588-590) ), but the current discussion is largely positioned on national policy and frameworks. Globally, organizations such as The mission to Seafarers, International Seafarers’ Wellness and Assistance Network, and the International Transportation Workers’ Federation, among others, have created wellness directories, multilingual helplines etc. The larger global mental health community will benefit from researchers commenting on how and what this body of research adds to it. Current implications section is focused on National level impact on policy and practice.

6. PLOS authors have the option to publish the peer review history of their article (what does this mean?). If published, this will include your full peer review and any attached files.

**Do you want your identity to be public for this peer review?** For information about this choice, including consent withdrawal, please see our Privacy Policy.

Reviewer #1: **Yes:** saumil dholakia

Figure Resubmissions:

---

## [Decision Letter · Decision Letter 1]

1 May 2026

Navigating mental health needs at sea: Perspectives from the Canadian maritime community during the COVID-19 pandemic

PGPH-D-25-03416R1

Dear Dr Rojas Aldieri,

We are pleased to inform you that your manuscript 'Navigating mental health needs at sea: Perspectives from the Canadian maritime community during the COVID-19 pandemic' has been provisionally accepted for publication in PLOS Global Public Health.

Best regards,

Susmita Chandramouleeshwaran

Academic Editor

Reviewer Comments (if any, and for reference):

Reviewer's Responses to Questions

**Comments to the Author**

1. If the authors have adequately addressed your comments raised in a previous round of review and you feel that this manuscript is now acceptable for publication, you may indicate that here to bypass the “Comments to the Author” section, enter your conflict of interest statement in the “Confidential to Editor” section, and submit your "Accept" recommendation.

Reviewer #1: All comments have been addressed

2. Does this manuscript meet PLOS Global Public Health’s publication criteria? Is the manuscript technically sound, and do the data support the conclusions? The manuscript must describe methodologically and ethically rigorous research with conclusions that are appropriately drawn based on the data presented.

Reviewer #1: Yes

3. Has the statistical analysis been performed appropriately and rigorously?

Reviewer #1: Yes

4. Have the authors made all data underlying the findings in their manuscript fully available (please refer to the Data Availability Statement at the start of the manuscript PDF file)?

Reviewer #1: Yes

5. Is the manuscript presented in an intelligible fashion and written in standard English?

Reviewer #1: Yes

6. Review Comments to the Author

Reviewer #1: Thank you for addressing all the issues.

7. PLOS authors have the option to publish the peer review history of their article (what does this mean?). If published, this will include your full peer review and any attached files.

**Do you want your identity to be public for this peer review?** For information about this choice, including consent withdrawal, please see our Privacy Policy.

Reviewer #1: **Yes:** saumil dholakia
